# Multiple Venous and Pulmonary Artery Thrombosis as the Presenting Features of Spontaneously Reversible Nephrotic Syndrome after Exposure to SARS-CoV-2 Virus (Pfizer/BioNTech BNT162b2) Vaccination

**DOI:** 10.3390/vaccines10111888

**Published:** 2022-11-09

**Authors:** Theerachai Thammathiwat, Athiphat Banjongjit, Piyatida Chumnumsiriwath, Laor Chompuk, Apichaya Sripariwuth, Sutatip Pongcharoen, Talerngsak Kanjanabuch

**Affiliations:** 1Department of Medicine, Faculty of Medicine, Naresuan University, Phitsanulok 65000, Thailand; 2Nephrology Unit, Department of Medicine, Vichaiyut Hospital, Bangkok 10400, Thailand; 3Department of Pathology, Faculty of Medicine, Naresuan University, Phitsanulok 65000, Thailand; 4Department of Radiology, Faculty of Medicine, Naresuan University, Phitsanulok 65000, Thailand; 5Division of Nephrology, Department of Medicine, Faculty of Medicine, Chulalongkorn University, Bangkok 10330, Thailand; 6Center of Excellence in Kidney Metabolic Disorders, Faculty of Medicine, Chulalongkorn University, Bangkok 10330, Thailand

**Keywords:** BNT162b2, COVID-19, minimal change disease, SARS-CoV-2, venous thromboembolism

## Abstract

We report an unusual case of nephrotic syndrome and multiple venous thromboembolism (VTE) four days after BNT162b2 injection. The patient presented with a three-day history of foamy urine and one-day history of right leg swelling. The investigation showed 9.5 g of 24 hr urine protein, hypoalbuminemia (2.6 gm/dL), and hypercholesterolemia (320 mg/dL). The duplex ultrasonography revealed VTE of the right lower extremity veins (common femoral vein, saphenous vein, and popliteal vein). Computed tomography (CT) showed thrombosis of the infrarenal inferior vena cava (IVC) extending to both iliac veins and acute pulmonary embolism. Kidney biopsy was performed. The diagnosis of minimal change disease was made. The patient received anticoagulation without steroid or immunosuppressive medications. The nephrosis was spontaneously resolved in 20 days; thus, it strongly suggested the diagnosis of vaccine-induced minimal change nephropathy. Reports of kidney adverse events and clinical characteristics are further needed in the circumstances of worldwide SARS-CoV-2 vaccine usage.

## 1. Background

Severe Acute Respiratory Syndrome Coronavirus 2 (SARS-CoV-2) vaccines have been widely used. In the light of the Omicron wave, Centers for Disease Control and Prevention (CDC) recommended using messenger ribonucleic acid (mRNA) vaccines BNT162b2 or mRNA-1273 [1]. Eight cases of de novo minimal change disease after BNT162b2 injection have been reported at the time of this report’s writing [2,3,4,5,6,7,8]. Patients’ age was in the range of 14–80 years old. Almost all patients had symptoms in the first week after the vaccinations and were in remission after receiving high-dose steroids. No patient developed venous thromboembolism (VTE). In this article, we report a case of spontaneously recovered minimal change disease (MCD) after BNT162b2 injection, complicated by multiple VTE and pulmonary artery embolism.

## 2. Case Presentation

A 17-year-old man presented with three-day foamy urine and one-day right leg swelling and low-grade fever. He had received the first dose of mRNA SARS-CoV-2 vaccine (Pfizer/BioNTech BNT162b2) four days prior. Physical examination revealed non-pitting edema with a positive Homan sign on his right leg. He denied past medical illnesses, familial history of kidney and VTE, smoking, drug, and alcohol abuse. Initial laboratory revealed massive proteinuria (urine dipstick 4+ and 9.5 gm of 24-h urine protein), hypoalbuminemia (2.6 gm/dL), hypercholesterolemia (320 mg/dL), and lipiduria (positive oval fat bodies and fatty cats). His serum creatinine was 0.72 mg/dL. His nasal swab for SARS-CoV-2 by Real Time Polymerase Chain Reaction (RT-PCR) was negative. Two days after admission, the patient developed dyspnea without desaturation. A respiratory physical examination revealed normal and equal breath sounds. Chest X-ray was unremarkable. D-dimer was positive (>40,000 ng/mL). Duplex ultrasound demonstrated venous thrombosis of the right lower extremity veins (common femoral vein, saphenous vein, and popliteal vein). Computed tomography (CT) venography of the abdomen revealed thrombosis of the infrarenal inferior vena cava (IVC) extending to both iliac veins (Figure 1A) while CT pulmonary angiography demonstrated acute pulmonary embolism of the right main pulmonary artery, confirming a diagnosis of pulmonary embolism (Figure 1B). Low molecular weight heparin (enoxaparin at a dosage of 0.6 mL twice a day) was promptly started and was interrupted for 24 h for kidney biopsy a week later when the patient’s conditions of multiple thromboses were stable (e.g., no desaturation, unchanged left leg diameter). The kidney biopsy was performed revealing normal 18 glomeruli (Figure 1C) and tubulointerstitial structures with all immunofluorescent studies being negative, including IgM, IgG, IgA, C3, C1q, Kappa, and Lambda immunoglobulin light chain. Electron microscopy revealed diffused podocyte foot process effacement and microvillous transformation without electron dense deposit. The provisional diagnosis was MCD. To exclude secondary illnesses causing MCD, viral hepatitis profiles, treponemal, antinuclear, and anti-HIV antibodies were screened, and all turned out to be negative. According to VTE, the secondary causes other than MCD were also examined. His hemoglobin was 17.4 g/dL with negative JAK2 V617F mutation and a normal erythropoietin level of 20.5 mIU/mL (reference range 3.7–29.5 mIU/mL). Prothrombotic factors were measured; homocysteine 8.5 µmol/L (reference range 5–15 µmol/L), negative antiphospholipid antibodies (anti-beta 2 glycoprotein I IgM was 25 RU/mL at admission and 13 RU/mL at 12-week follow-up (reference range 0–20 RU/mL); inconclusive lupus anticoagulant at admission and negative at 12-week follow-up; and negative anti-cardiolipin IgM/IgG). Antithrombotic factors were also performed: antithrombin III activity 93% (reference range 79–112%), protein C activity 71% (reference range 70–140%), and protein S activity 109% (reference range 60–150%). The platelet factor 4 antibodies tested by ELISA-based assay revealed a negative result. The inflammatory marker of C-reactive protein (CRP) was 17.65 mg/L (reference range 0.3–5.0 mg/L). The complement levels were elevated: C3 248 mg/dL (reference range 100–233 mg/dL), C4 74 mg/dL (reference range 14–48 mg/dL), and CH50 > 51 U/mL (reference range 20–40 U/mL). One day after kidney biopsy, enoxaparin was resumed and prophylactic therapy with warfarin at dosage 7.5 mg daily was started. INR ranging 2.0–3.0 folds were maintained. Nephrotic syndrome subsided without any corticosteroids or immunosuppressive medications. 20 days post-admission, the patient’s right leg circumference gradually decreased, and the proteinuria significantly declined (0.3 gm/day) (Figure 2). The complete remission of MCD remained until a year after the onset of symptoms; there was no clinical or laboratory recurrence of MCD or clinically extensive thrombosis. The booster of SARS-CoV-2 vaccination was refused by the patient without subsequent experience of SARS-CoV-2 infection.

## 3. Discussion 

SARS-CoV-2 vaccines, especially mRNA vaccines, can reduce the symptomatic infection and severe infection of the Omicron strain of Coronavirus disease (COVID-19) [9]. The common adverse events of BNT162b2 injection include fatigue, headache, muscle pain, and fever [10]. However, rare kidney adverse events, including interstitial nephritis and glomerular diseases were sporadically observed, although a causal relationship could not be inferred. Reported glomerular diseases consist of MCD (most common), followed by membranous nephropathy, IgA nephropathy, anti-glomerular basement membrane nephritis, and antineutrophil cytoplasmic antibodies (ANCA) associated vasculitis [11]. MCD was hypothesized to be a result of a prevalence of circulating CD8+ T suppressor cells, type 2 T helper cells (Th2), and cytokines produced from Th2 (interleukin (IL) 4, IL8, and IL13) that can injure the foot processes of podocytes [12,13]. A reduced function of regulatory T cells is also a proposed mechanism. The secondary causes of MCD involve allergy, malignancies, some drugs, infections, and autoimmune disorders [13]. After SARS-CoV-2 infection, SARS-CoV-2-specific T cells can be detected in blood circulation 2–4 days after the onset of symptoms [14]. This could infer that T cells are activated within one week after the SARS-CoV-2 vaccination. It would be of interest if the cytokines were consecutively measured during the cause of MCD to confirm this causal relationship; unfortunately, the investigations were not performed. However, an elevation of non-specific systemic inflammatory response parameters, including CRP and complement levels [15], was demonstrated in this patient, possibly indicating immune activation following BNT162b2 vaccination. Although a causal relationship between vaccines and the disease cannot be concluded, the temporal association between administration of vaccination and the onset of nephrotic syndrome supports the relationship as documented by the presented case and reports from other groups [2,3,4,5,6,7,8]. Additionally, the fact that all potential secondary causes of MCD were excluded, together with the spontaneous resolution of the disease without steroid treatment, strongly supports the causal relationship between the vaccine and MCD.

The first-line treatment of de novo minimal change disease is high-dose prednisolone with a dosage of 1 mg/kg/day (maximum 80 mg/day) for a minimum of 4 weeks, tapered after remission over at least 24 weeks [16]. Fifty percent of patients respond to 4 week glucocorticoid and increase by 10%–25% at 16 weeks [16]. The treatment of secondary MCD other than eliminating the cause is not well established. Nonsteroidal anti-inflammatory drug (NSAID)-associated MCD can be spontaneously resolved after drug discontinuation without steroid treatment [17]. In all previous reports of MCD after BNT162b2 injection, the patients received high-dose steroids, and most achieved partial or complete remission. In this case, we reported spontaneously resolved BNT162b2-related MCD without steroid treatment. Spontaneous MCD remission had been reported in a patient developing MCD after influenza B infection which could be presumed, similar to our case, to be a result of the normalization of the imbalanced immune system after the resolution of the infection. Although younger MCD patients can achieve earlier remission with corticosteroids [18], the correlation of age and spontaneous remission of idiopathic MCD has been rarely evaluated, since corticosteroids are the cornerstone of the treatment; thus, almost patients received early corticosteroids [19].

VTE is the second most life-threatening complication, after the infections, of nephrotic syndrome [20]. Recent meta-analysis showed that the overall prevalence of pulmonary thromboembolism in nephrotic syndrome patients was 8% (95% confidence interval [CI] 4.27–14.73) [21]. The main risk factors were male gender, aging, massive proteinuria (proteinuria > 8 gm/day increase the risk of VTE by 17 folds), low serum albumin (serum albumin < 2.5 gm/L increased the risk of VTE three-fold; proteinuria to serum albumin > 5 increased the risk of VTE six-fold). The most common type of glomerulopathy-causing VTE is membranous nephropathy [20]. Kidney Disease: Improving Global Outcomes (KDIGO) 2021 Clinical Practice Guideline for the Management of Glomerular Diseases recommends anticoagulant prophylaxis, considering together with bleeding risk if patients have serum albumin < 2.0–2.5 gm/L and any of the following: proteinuria > 10 g/d, body mass index > 35 kg/m^2^, genetic disposition for thromboembolism, heart failure New York Heart Association class III or IV, recent orthopedic or abdominal surgery, or prolonged immobilization [16]; this was not applicable to our patient, hence the patient did not receive prophylactic anticoagulant, despite their high VTE risk according to the aforementioned meta-analysis. Fenton et al. reported that 12% of 78 adult MCD patients developed VTE and 78% of them had VTE at the time of nephrotic syndrome [22]. The mechanisms of VTE in nephrotic syndrome patients are urinary loss of anticoagulant proteins (antithrombin III, protein C, and protein S) and increased synthesis of prothrombotic factors which are acute-phase reactants (factors V and VIII, von Willebrand factor, fibrinogen, and α2-macroglobulin) [23]. In our patient, we found no abnormal measurable anticoagulant proteins; thus, the VTE was assumed to be caused by increased unmeasured prothrombotic factors.

## 4. Conclusions

Herein, we presented a case of BNT162b2-related minimal change disease which was spontaneously resolved but complicated by multiple venous thrombosis and pulmonary embolism. Reports of kidney adverse events and clinical characteristics are further needed in the circumstances of worldwide SARS-CoV-2 vaccine usage.

## Figures and Tables

**Figure 1 vaccines-10-01888-f001:**
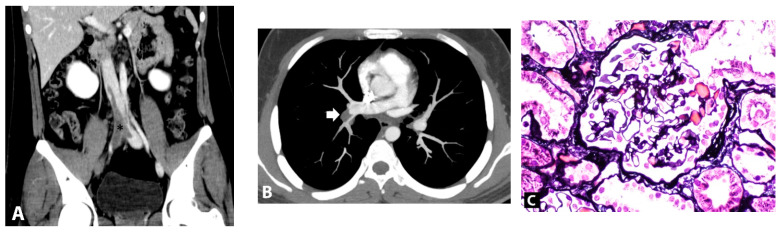
(**A**)**.** Computed tomography showed partial thrombosis of the inferior vena cava and total occlusion of right iliac veins (asterisk). (**B**). Axial computed tomography pulmonary angiography image showed pulmonary embolism at the right descending pulmonary artery (arrow). (**C**). Kidney biopsy finding, light microscopy of glomerulus with Jone silver stain (original magnification, ×200) revealed normal glomerulus.

**Figure 2 vaccines-10-01888-f002:**
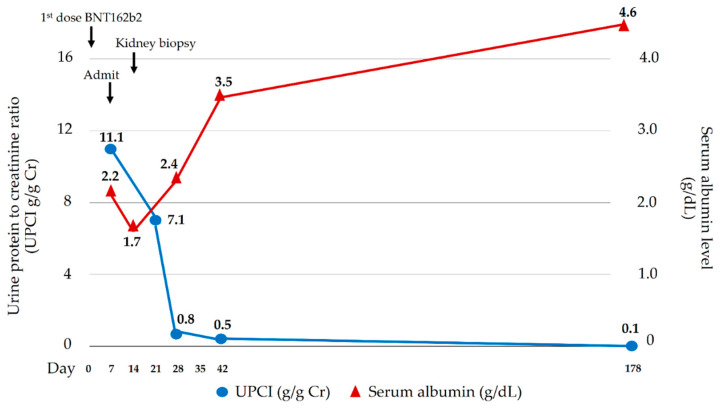
Clinical courses of nephrotic syndrome.

## Data Availability

Not applicable.

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
