# Peer review of "Multiple Venous and Pulmonary Artery Thrombosis as the Presenting Features of Spontaneously Reversible Nephrotic Syndrome after Exposure to SARS-CoV-2 Virus (Pfizer/BioNTech BNT162b2) Vaccination"

_vaccines, 2022, doi:10.3390/vaccines10111888_

Round 1
Reviewer 1 Report
In the case report “Multiple venous and pulmonary artery thrombosis as the presenting features of spontaneously reversible nephrotic syndrome after exposure with SARS-CoV-2 virus (Pfizer/BioNTech 4 BNT162b2) vaccination” the authors reported a case of spontaneous recovery from minimal change disease after BNT162b2 injection complicated by multiple venous thromboembolism (VTE) and pulmonary artery embolism in 17-year-old men with no past medical and family history of kidney diseases and VTE. By presented data, it could be assumed that MCD was induced by BNT162b2.
Comments:
1. In the Figure 1A and 1B, please put arrows to indicate the change!
2. Was any cytokine level measured in plasma of the patient, especially Th2 cytokines (IL-4, IL-5, IL-13 among others)? That would be important to know and will support discussion of the case report.
3. Page 2, line 60: “his clinical of multiple thromboses was stable”; after clinical it should be added symptoms or signs.
4. English should be checked and improved throughout the report.
Author Response
In the case report “Multiple venous and pulmonary artery thrombosis as the presenting features of spontaneously reversible nephrotic syndrome after exposure with SARS-CoV-2 virus (Pfizer/BioNTech 4 BNT162b2) vaccination” the authors reported a case of spontaneous recovery from minimal change disease after BNT162b2 injection complicated by multiple venous thromboembolism (VTE) and pulmonary artery embolism in 17-year-old men with no past medical and family history of kidney diseases and VTE. By presented data, it could be assumed that MCD was induced by BNT162b2.
Comments:
- In the Figure 1A and 1B, please put arrows to indicate the change!
Reply: Thank you for your recommendation. We put the asterisk and arrow in the images as your suggestion.
- Was any cytokine level measured in plasma of the patient, especially Th2 cytokines (IL-4, IL-5, IL-13 among others)? That would be important to know and will support discussion of the case report.
Reply: We did not performed the test. Thus, we add this suggestion in the discussion.
“It would be of interest if the cytokines were consecutively measured during the cause of MCD to confirm this causal relationship; unfortunately, the investigations were not performed.”
- Page 2, line 60: “his clinical of multiple thromboses was stable”; after clinical it should be added symptoms or signs.
Reply: Thank you. We changed to “the patient’s conditions of multiple thromboses were stable”
- English should be checked and improved throughout the report.
Reply: Thank you. We performed an English check.
Reviewer 2 Report
Dear authors
Your valuable case report here described the pathological situation of the case very well but I would have preferred if you fixed as well on the immunological state after vaccination to link the MCD & VTE. In lines 97,98&99 you suggested that ‘ MCD was hypothesized to be a result of a prevalence of circulating CD8+ T suppressor cells, type 2 T helper cells (Th2), and cytokines produced from Th2 (interleukin 13)‘. I would have preferred that you have tested that hypothesis by assessing the levels of these cells together with interleukin 13, please clarify why.
On the other hand you have managed the case successfully without immune suppression. Please include a clarification of that in your discussion. There is also the young age of the patient that of course could have helped in the reversal of his MCD, please include that in our discussion of previous cases. Thank you again for reporting your successful case.
Author Response
Your valuable case report here described the pathological situation of the case very well but I would have preferred if you fixed as well on the immunological state after vaccination to link the MCD & VTE. In lines 97,98&99 you suggested that ‘ MCD was hypothesized to be a result of a prevalence of circulating CD8+ T suppressor cells, type 2 T helper cells (Th2), and cytokines produced from Th2 (interleukin 13)‘. I would have preferred that you have tested that hypothesis by assessing the levels of these cells together with interleukin 13, please clarify why.
Reply: We did the descriptive review of the mechanism of MCD in generalized cases. In this case, we did not measure the cytokine levels. Thus, we added this recommendation in the discussion.
On the other hand you have managed the case successfully without immune suppression. Please include a clarification of that in your discussion. There is also the young age of the patient that of course could have helped in the reversal of his MCD, please include that in our discussion of previous cases. Thank you again for reporting your successful case.
Reply: Thank you for your recommendations. We added the discussion regard to the age and spontaneous remission.